# Development of a Novel Modular Compliant Gripper for Manipulation of Micro Objects

**DOI:** 10.3390/mi10050313

**Published:** 2019-05-09

**Authors:** Matthew Lofroth, Ebubekir Avci

**Affiliations:** School of Food and Advanced Technology, Massey University, Palmerston North 4410, New Zealand; ma.lofroth@gmail.com

**Keywords:** micro mechanisms, micro manipulation, cell manipulation, compliant micro gripper, piezo actuator

## Abstract

This paper proposes a modular gripping mechanism for the manipulation of multiple objects. The proposed micro gripper combines traditional machining techniques with MEMS technologies to produce a modular mechanism consisting of a sturdy, compliant aluminium base and replaceable end-effectors. This creates an easily-customisable solution for micro manipulation with an array of different micro tips for different applications. We have provided the kinematic analysis for the gripper to predict the output and have also optimised design parameters based on FEA (finite element analysis) simulation and the effects of altering flexure beam lengths. The gripper is operated by a piezo actuator capable of 18 μm displacement at 150 V of applied DC voltage. This is then amplified by a factor of 8.1 to a maximum tip displacement of 154 μm. This is achieved by incorporating bridge and lever amplifying techniques into the design. An initial experimental analysis of the micro gripper is provided to investigate the performance of the micro gripper and to gauge the accuracy of the theory and simulation through comparison with experimental results.

## 1. Introduction

“Nano technology. The science is good, the engineering is feasible, the paths of approach are many, the consequences are revolutionary-times-revolutionary, and the schedule is: in our lifetimes” said Stewart Brand in the foreword of the book “Unbounding the Future: The Nano Technology Revolution” [1]. Modern engineering has allowed us to step into this bizarre world of micro- and nano-technology, which has advanced research in biomedical applications [2], micro manipulation [3], micro assembly [4] and many other areas.

The field of manipulating micro objects simply and effectively has been widely discussed in recent literature. Having the ability to perform experiments on single-cell organisms is crucial as each cell holds its own individual characteristics; meaning that significant heterogeneity can exist among cells. These differences can be due to many reasons including: localised damage, mutations, stages in cell life, differences in exposure to external signals and many other reasons [5]. For example, brain cells may express as few as 65% of the same genes as their neighbours [6].

One solution to the problem of single-cell manipulation is known as “optical tweezers”, which involves no physical contact with an object. This is done by trapping a particle inside a beam of light using the change in momentum of photons [7]. Other non-contact manipulation techniques can include sound wave [8] and electrostatic [9] technologies. However, the majority of these methods are limited to single plane motion. Non-contact techniques can also damage the object with excessive heat and high current flows [10]. Due to the aforementioned problems, many studies have proposed contact manipulation techniques when working with living cells for many reasons, such as selectivity and compatibility [11,12,13,14]. There are still a few problems that we can identify with the current contact manipulation methods. The key problems can be listed as: adhesion due to force scaling laws, lack of easy replacement and modularity among gripping tips and, lastly, complex micro structures leading to fragility and a large amount of actuators/joints and fixtures.

The first problem exists due to force scaling laws, which means adhesion forces can make the release of an object extremely difficult [15]. Biological cells are generally sized between 1 and 100 μm with the majority of cells nearer to 10 μm. At this scale, adhesion forces become a problem as objects begin adhering to each other and are difficult to separate. Because of this, many studies have developed manipulation methods for relatively large-scale objects (100–300 μm), thereby avoiding the problem of adhesion forces [16].

Less common studies have shown micro grippers capable of single-cell manipulation [17]; however, this leads into our next problem. Hao et al. developed an electro-statically-driven gripper capable of handling objects within 20–60 μm [18]. The design was exclusively manufactured at a micro scale, which minimizes adhesion problems and also includes complex features such as a comb drive structure, delicate flexure beams and a ratchet self-locking system. The issue is that many of these systems in the literature capable of cell-sized object manipulation are designed in this way, and although this technique is an excellent display of MEMS technology, it leaves the entire structure extremely fragile and costly to replace. Replacement is often necessary as gripper tips can become polluted with unwanted particles or to avoid cross-contamination between samples. Many micro objects may also require gripper tips of different shapes and sizes, which further highlights the need for an easily-replaceable design.

The last problem with contact manipulation lies with the inherent difficulties of manufacturing micro mechanical joints and fixtures. This leads to the majority of micro grippers being under-actuated and/or compliant, which minimizes the need for many actuators and also means that the gripper tips are more flexible for grasping asymmetrical objects [19]. A three-finger micro gripper capable of 10–800 μm-sized object manipulation was recently proposed by Tao Chen et al. and explored the idea of replaceable gripper tips [20]. Although each tip could be changed, proper alignment of the three gripping fingers required multiple actuators and compliant hinges, which adds unwanted bulk, cost and complexity to the system. Another example of a replaceable micro gripper proposed that each gripping tip could be glued to an end effector probe by using a UV resin [21]. The main issue with this method is that gluing the tips requires a highly-precise process and also takes roughly 10 min to complete. From these examples, it is clear that a good gripper design should not only be replaceable, but should also be simple enough that gripping tips can be changed efficiently with little effort.

Compliant designs offer other benefits such as monolithic manufacturing and repeatable motion through elastic deformation of structural members, meaning there are no friction and backlash involved [22]. There are many ways of actuating such compliant mechanisms such as shape memory alloys [23], electro-thermal actuators [24], electromagnetic [25] and electrostatic control [26]. Piezoelectric (PZT) actuators are the most common actuation choice due to their precise sub-nanometre positioning capabilities [27], high force output to weight ratio [28], fast response time and good performance with the proper control.

In this paper, we propose a piezo actuator-driven monolithic compliant micro gripper for the purpose of manipulating both single biological cells and standard micro particles sized between 6 μm and 500 μm for study. Currently, micro grippers capable of this are either highly fragile and costly micro structures or overly complex with multiple actuators. Firstly, we propose to minimize the complexity of our design by creating a large and strong compliant mechanism controlled by a single piezo actuator. Secondly, our gripper will offer replaceable gripping tips manufactured through a deep reactive ion etching process (DRIE), which can be easily attached to the larger mechanism. Lastly, we plan to improve the usability of the gripper by having custom gripper tips available for separate applications (Figure 1). For example, smaller objects would require finer tips for reducing adhesion forces or concave tips could offer a more firm grip around an object. By incorporating these three elements into our design, the cost of micro manipulation systems will be reduced while increasing the versatility of our gripper across a much larger range of micro objects/sizes than previously seen in literature. The following sections of this paper will include the materials and methods involved in the design of our system, followed by optimisation techniques, control algorithms and micro manipulation experiments and results.

## 2. Materials and Methods

### 2.1. Kinematic Modelling of the Micro Gripper

The output of the micro gripper x(t) (Figure 2) can be modelled through a set of geometric relationships based on the initial state of the gripper [29]. Two key displacement amplification techniques were used in the design of the gripper: firstly, a bridge amplifier [30], which converts horizontal motion to vertical motion by deflecting two parallel beams [31] and, secondly, a simple lever to translate the motion to the gripper tips. Because of this, the modelling has been split into two sections, firstly the bridge amplifier with initial design constants $u_{0},$
$L_{0},$
*w* and secondly the upper compliant section with a,
b,
$h_{0}$ and $L_{0}^{I}$ as defined in Table 1. These constants were optimised by comparing the gripper output based on the distance between link lengths. Furthermore, some of the constants were calculated to allow both angles $\beta_{0}$ to remain equal during the flexing of the gripper, which simplified our modelling equations (Figure 2).

Starting at the bridge amplifier, the piezo actuator applies a known horizontal input displacement of y(t) to the system. This causes the structure to flex, which decreases the angle $\theta_{1}$ and subsequently alters the length of $L_{0}$, yielding $L_{1}$. The output displacement h(t) of the bridge amplifier can therefore be found by taking the difference between the initial ($L_{0}$) and current ($L_{1}$) lengths (Equation (Equation 1)). This is then multiplied by 2 as the bottom end of the bridge is fixed and thus transfers its motion to the top.
(1)$$h\left(t\right)=\left|2*(L_{0}-L_{1})\right|$$
where:(2)$$L_{1}=\left(u_{1}\right)tan\left(\Theta_{1}\right),\;\;\;\;\;\;\;\Theta_{1}=cos^{-}1\left(\frac{u_{1}}{w}\right),\;\;\;\;\;\;\;u_{1}=u_{0}+y\left(t\right)$$

The horizontal displacement h(t) is then forwarded into the upper compliant section where it causes the length $h_{0}$ to decrease, creating a new length $h_{1}$. This subsequently alters angles $a_{0},$
$a_{0}^{I},$
$a_{0}^{II},$
$b_{0},$
$b_{0}^{I},$
$b_{0}^{II}$ and $\beta_{0}$, yielding $a_{1},$
$a_{1}^{I},$
$a_{1}^{II},$
$b_{1},$
$b_{1}^{I},$
$b_{1}^{II}$ and $\beta_{1}$. By using the micro gripper constants in combination with these changes, the output x(t) (Equation (Equation 3)) can be calculated as:
(3)$$x\left(t\right)=\left(L_{1}^{II}+h\left(t\right)\right)\;\tan\left(\beta_{1}\right)$$
where: (4)$$b_{1}=cos^{-1}\left(\frac{a^{2}+c_{1}^{2}-b^{2}}{2ac_{1}}\right),\;\;\;\;\;\;\;\;\beta_{1}=\frac{\pi }{2}-a_{1}^{I}$$

(5)$$c_{1}=\sqrt{{\left(h_{1}\right)}^{2}+{\left(L_{1}^{I}\right)}^{2}},\;\;\;\;\;\;\;\;\gamma_{1}=\pi -b_{1}-a_{1}$$

(6)$$a_{1}^{I}=a_{1}+a_{1}^{II},\;\;\;\;\;\;\;\;a_{1}^{II}=b_{1}^{II}=tan^{-1}\left(\frac{h_{1}}{L_{1}^{I}}\right)$$

(7)$$a_{1}=cos^{-1}\left(\frac{b^{2}+c_{1}^{2}-a^{2}}{2bc_{1}}\right),\;\;\;\;\;\;\;\;h_{1}=h_{0}-h\left(t\right)$$

For the purpose of the kinematic analysis, a pseudo-rigid body model (PRBM) of the micro gripper was created. This displays design variables and geometric relationships associated with the mechanism (Figure 2). This method is commonly used among compliant architectures, as it simplifies analysis by assuming flexure hinges as being torsional springs rather than rotational hinges. This model provided a method for calculating the workspace of the gripper based on an input to the piezo actuator. The results from our model are very similar to the measured experimental actuation of the gripper when viewed side by side (Figure 3).

### 2.2. Compliant Gripper Fabrication

The micro gripper base was machined from a 7075-T6 aluminium plate. The strength and elasticity of this material make it a common choice for compliant mechanisms [32]. The micro gripper was actuated by a resin-coated multilayer piezo actuator capable of approximately 19 ± 2 μm total expansion within 0–150 volts. This provided a maximum blocking force of 1700 N and was pre-tensioned by a small screw in the bridge amplifier. Two sections of the aluminium structure were designed to amplify this motion by approximately 8.1-times to the gripper tips. This achieved a large range of output motion with a maximum stroke of 154 μm. The first section utilized a compliant bridge mechanism that was able to convert horizontal motion from the actuator to vertical displacement in the end-effectors (Figure 4). This motion was then further amplified and translated to the micro-gripper tips by the use of a lever mechanism. The gripper structure was designed symmetrically along the centre vertical axis in order to achieve a similar, but opposite motion from both actuating fingers and also avoid shear and bending forces acting on the actuator.

### 2.3. Micro Tip Fabrication

#### 2.3.1. Silicon Tip Fabrication

As mentioned previously, the gripper was designed to have a variety of interchangeable tips for different applications. These tips were manufactured from 400 μm-thick silicon wafers by using a DRIE (deep reactive ion etching) approach. The different stages involved for manufacturing a set of tips are shown in Figure 5. Stage 1 began with the preparation of a silicon wafer. In order to remove any impurities from the wafer surface, each wafer was first ultrasonically cleaned in acetone for 10 min. This was followed by a 10-min plasma clean to remove any remaining organic particles. Once thoroughly cleaned, we then added a 500 nm-thick layer of chromium to each wafer (Stage 2). This was done by using electron beam sputtering technologies (BOC EDWARDS FL400), which used a powerful beam of electrons to vaporise and deposit the chrome evenly on each wafer. The third stage involved spinning a 1-μm layer of photo resist onto each wafer, where the etching pattern would eventually be placed. Once complete, a mask that contained the etching pattern needed to be created. This was done by using CAD software to create a 2D image of the pattern, which was then loaded into the mask writer (Heidelberg UPG 101). The mask writer would then write the pattern onto a blank mask, which was comprised of three layers; glass, chromium and photo resist. Once complete, the mask was then developed in developing liquid to reveal the inverted pattern of the grippers. Chrome etching liquid was then used to remove the unwanted chrome around the mask, leaving only transparent glass and the micro gripper etching pattern. We could now move onto mask alignment (Stage 4), where we placed the mask and wafer into the alignment device (Karl Suss MA6). After aligning the pattern over the wafer, the MA6 then delivered a powerful 10-second burst of UV light over the mask, subsequently transferring the pattern to the wafer photo resist. A quick 1-minute bath in developing liquid then exposed the desired pattern on the wafers. The fifth stage involved first bathing each wafer in chrome etching solution to remove unwanted chrome around the exposed pattern. We then repeated ultrasonic cleaning in acetone to remove the photo resist, which was no longer needed, leaving only the chrome etching pattern and silicon. The reason why we created the pattern in chrome was that the DRIE process etched through chrome much slower than silicon. The idea was that the chrome would protect the silicon from etching in unwanted areas. At this stage, we could move to Stage 6 by placing the wafer into the DRIE device (Oxford Instruments PlasmaPro100 Cobra) where the unwanted chrome and silicon would be etched away, leaving us at Stage 7, where only the chrome-covered gripper tips remain. The remaining chrome was removed through wet etching in the last stage (8), leaving pure silicon gripper tips.

#### 2.3.2. Brass Tip Fabrication

In order for the silicon micro gripper tips to be correctly aligned, they were manufactured as a single part. This means the tips must flex with the gripper; however, silicon was found to be too brittle for any compliant structure to be included. Because of this, more flexible 0.3 mm-thick brass tips were created with a custom made micro mill upon which the silicon tips would be attached. This was done simply by using Loctite glue, and because the brass and silicon tips were dimensionally very similar, a surface tension phenomena caused them to align naturally upon gluing [33]. To keep the silicon tips as a single part, a small beam connected the tips while manufacturing, which made them rigid. This beam was then broken once the tips were glued to the brass, allowing them to flex with the compliant brass tips (Figure 6).

### 2.4. Gripper Tip Alignment

Achieving near perfect alignment (within 1–2 μm) of the micro gripper tips is crucial for stable manipulation of small spherical objects, such as micro beads or cells. If the tips are not applying a force normal to the equator of the micro object, it is very difficult to adequately grasp it, such that the object can be raised and moved from the substrate floor (Figure 7)

By creating the silicon gripper tips as a single part, they were naturally aligned in all x, y, and z directions. As mentioned previously, one of the research aims for this micro gripper was to reduce the amount of micro fabrication required to create a replaceable, simple and yet capable design. This means that the design tolerances for each part (compliant aluminium base, brass and silicon tips) were vastly different, with the silicon tips being the only part created with micro-scale features. Because of this, alignment of the silicon tips in the z-axis (up/down direction) was lost when all parts were assembled. This is due to slight flexing of the tips when attached to the aluminium and brass parts, as they were not manufactured with the same level of precision as the silicon tips and were not aligned at the micro-scale. This also means that when assembling the gripper, the silicon micro tips will always have a slightly different orientation relative to the compliant base when assembled. This however was expected and is not a problem as the large compliant base will always apply a normal force to each side of the silicon gripper tips, regardless of slight differences in alignment. These small changes in alignment were dealt with by a simple solution that made use of extremely fine M2.5 × 0.20 adjustment screws (Thorlabs F2D5ES8). When turned, the screws make contact with the gripper tips, causing them to flex slightly downwards in the z-axis, meaning that alignment inconsistencies due to different manufacturing tolerances are easily corrected (Figure 8).

Glass end effectors can also be attached to the gripper, which were created with a Narishige PC-100 pulling device, which heated, softened and stretched a glass rod. When using glass end effectors, x-axis alignment (left/right) must also be considered as each end effector was fixed separately to the compliant base. For Z-axis alignment, a single fine adjustment screw was used to press down on each end effector slightly (Figure 9). Another fine screw was also used to press against the back side of the end effector. This meant that when tightened/loosened, both z-axis (up/down) and x-axis (forward/backward) alignment was now possible.

### 2.5. System Configuration

In order to manipulate a sample using the micro gripper properly, very precise motorized micro stages were required. In this study, a set of 3-axis motorized stages (OSMS80 and HPS80, Sigma Koki) with a controllable resolution of 0.5 μm was used. These stages were controlled serially by a driver (SHOT304, Sigma Koki) from a PC (Figure 10). A custom-machined bracket was attached to this, to which the micro gripper was fastened. The gripper was actuated by a NEC/TOKIN AE505D18H18F resin-coated piezo actuator, which was driven by a Matsusada PZJ-0.15Px3 driver (Figure 11)

An ADS1258 16-bit DAC and 24-bit ADC were connected to a Raspberry Pi 3 in order to control the entire system remotely from the PC. This was done by using two Cytron 433 MHz radio UART transceiver modules, one of which was connected to the Pi and the other to the computer. The Pi could now communicate strain gauge data wirelessly to the computer for viewing and vice versa, and the PC could send commands to the Pi for operation of the gripper. A diagram describing this communication can be seen in Figure 12. The microscope used was an Olympus IX71 with a U-Eye 1540SE-M-GL microscope camera with a resolution of 1024 × 768. The magnification for experiments was set to 4× for silicon tips and 40× for glass end effectors with viewing areas of 1580 × 1253 μm (2 μm per pixel) and 246 × 195 μm (0.5 μm per pixel) respectively.

### 2.6. Gripper Control

In order to drive a piezo actuator, a stable high voltage of 0–150 VDC was required. For this, a Matsusada PZJ high-speed piezo driver was used to control the piezo with 0.1-V precision, which equated to 0.1 μm of motion at the tips. The driver was adjusted with a raspberry pi 3 Linux operating system with an ADS1258 attachment. The ADS1258 offers a 16-bit DAC, which we used to send an analogue voltage between 0 and 10 VDC to the piezo driver in order to adjust the high voltage.

Furthermore, a single 120-ohm Kyowa KFGS-6-120-C1 strain gauge was glued to the piezo actuator to get a reading of the expansion for closed-loop control. The gauges were connected to a simple Wheatstone half-bridge circuit, which measured changes in resistance as the strain gauges expanded. The idea of the Wheatstone half-bridge circuit was to balance the resistance of the bridge to approximately 120 ohm initially. When the bridge was balanced, approximately 0 VDC was read when measuring the voltage across the middle of the bridge. When the strain gauges were expanded, their resistance increased, effectively unbalancing the bridge. This caused the voltage measurement to increase slightly between 0 and 1.5 mVDC over the entire expansion of the piezo actuator (0–19 μm). In order to obtain any usable information from this small change, a 24-bit ADC (analogue-to-digital converter) was used to sample the data with high resolution. The ADS1258 was again used, as it also includes a 24-bit ADC, which provided a high precision measurement across the bridge with a resolution of micro volts. Closed-loop control was accomplished by comparing the current actuator strain with a set of previously-calibrated strain values and resulting expansions. The entire closed loop control can be visualised in Figure 13, where the input is a given gripping set point and the output is the expansion of the piezo actuator.

#### Object Grasping Process

Adequately gripping an object plays a large role in pick and place success. It is also important to avoid unwanted damage being caused to the object from excessive grasping force. The target object size is always changing; therefore, the gripper tips’ distance from the substrate floor must also change for each pick and place task. Our gripper employed a multi-stage gripping process to grasp an object safely. Once positioned around the object, the gripper was first lowered to touch the substrate floor. The gripper was then incrementally raised where contact with the object was attempted at each step. The third step involved visually confirming that the tips were contacting the object equator, as it would begin to move when touched. If the tips were either above or below the object, part of the tips would become obscured by the object before contact was made. Therefore, when the tips were fully visible and the target object was moving, the micro gripper was considered to be aligned with an object. The final step (grasping step) was the most important step to avoid causing damage to the object. In the stage, the tips were slowly closed in increments of 0.1 μm. The gripper was then slightly moved to confirm if the object was adequately grasped. Performing this step in such small increments allowed us to avoid applying large forces to the object by over grasping. Furthermore, by converting pixels into distance, we were able to confirm this movement by measuring how far the tips were travelling and, also, when an object began to move.

## 3. Experiments

### 3.1. Gripper Calibration

To calibrate the system, a set of data relating strain to gripper expansion was gathered with a simple experiment. A program was created to expand and retract the actuator automatically by incrementally adjusting the input voltage in steps of 1 V (0.12 μm). This was done to cover the entire range of the actuator while simultaneously recording the strain at each point. The results for two expand/retract cycles can be seen in Figure 14. The data revealed that there was considerable hysteresis when comparing the expansion and retraction of the actuator. Furthermore, a small amount of error was found to be present when comparing the strain results from different expand/retract cycles. In order to account for this, the actuator was expanded and retracted a total of 10 times with strain data being recorded. Each set of data was then averaged together to get a more precise calibration; however, when examining each test, very little difference was visible between the results. The graph in Figure 14 shows the results for two actuator cycles side by side. The data from these two tests showed the greatest amount of variance amongst all 10 tests. Even so, the largest error between the expansion and retraction cycles was found as to be 0.5% and 0.2%, respectively. When this error was converted to actuator expansion, the difference became 0.075 μm when expanding and 0.03 μm when retracting. When translated to the gripper tips, this error was amplified and became 0.61 μm and 0.24 μm for expanding and retracting, respectively. This error was extremely small and was considered negligible when manipulating objects within our size range (6 μm–500 μm).

Lastly, because the strain gauge was firmly glued to the piezo actuator, re-calibration was only required if the actuator was changed or if the strain gauge was roughly bumped/tugged. Although the compliant aluminium base may change in stiffness over time, the calibration would be unaffected due to the piezo actuator’s large blocking force (1700 N).

### 3.2. Shaped Gripper Tip Manipulation

A simple experiment was conducted to compare the functionality of four different silicon tip types when manipulating spherical and non-spherical micro beads (Figure 15).

#### 3.2.1. Method

The spherical and non-spherical micro beads used in this experiment were both made from glass and were approximately sized between 100 μm and 500 μm. In order to conduct experiments, a glass slide was first cleaned with isopropyl alcohol and placed on the microscope. A small sample of micro beads was then deposited onto the slide followed by the gripper being slowly lowered until adequate gripper-object alignment was achieved. Each tip was then used to perform a single trial of 20 pick and place tasks for both spherical and non-spherical beads. Each bead was randomly selected in terms of size (between 100 μm to 500 μm) and shape (for non-spherical only). The first task involved recording the total time for a user to pick and place the bead. When grasping a bead, the number of attempts before achieving a stable grip was also recorded. Although some results took more than one attempt to grip the bead, it was eventually grasped for all 20 takes.

#### 3.2.2. Results and Discussion

Figure 16 shows the average time taken to complete a successful micro bead pick and place task for all tip types. This was done to gauge the efficiency of using shaped tips on different target objects.

From these data, two main possibilities can be drawn for spherical object manipulation. Firstly, for spherical beads, round tips on average took the least average time (42.75 s) to complete the tasks. This was closely followed by toothed tips, which took an average time of 47.8 s. This result was expected, as a spherical bead tends to fit comfortably and firmly in between the concave curves of a round tip (Figure 17). It also suggests that for spherical target objects, a round tip shape is the most efficient for manipulation. Secondly, flat tips, on the other hand, seemed to show a considerable increase in average time (85.55 s). This was mostly due to the difficulties in grasping the target object, as the bead would often be uncontrollably flicked away. The sharp tips come in roughly somewhere between the results at 67.55 s.

When compared to non-spherical objects, the results clearly show that toothed tips are the most efficient with less than half the average time (34.35 s) of sharp, round and flat tips, which took 100.05, 93.6 and 97.8 s, respectively. This is most likely due to the toothed tips having multiple points of contact with an object, making it more suited for interlocking with an uneven surface.

The amount of attempts before having a firm grasp on each micro bead was recorded and plotted in Figure 18. This was done to highlight if a particular tip shape showed any signs of being easier to use when dealing with spherical and non-spherical beads.

The results show that the amount of grasping attempts closely followed the shape of the manipulation time graph. This is expected as more grasp attempts greatly increase the time taken. The data shows that round tips were not only the most time efficient, but were also the easiest to use when dealing with spherical objects. We can also see that toothed tips followed round tips very closely with an average attempt rate of 1.1 as opposed to 1.05. This difference is likely negligible and means round and toothed tips functioned very similarly when grasping spherical objects.

This case was different for non-spherical objects with toothed tips showing an attempt rate less than half of the round tips and four-times less than sharp and flat tips. This result, along with the small manipulation time, clearly suggests that toothed tips offered a vastly more efficient method for manipulating non-spherical objects.

### 3.3. Biological Object Manipulation

Manipulation of micro beads is a good method for showing the dexterity and functionality of a micro gripping system; however, in reality, there are many other types of micro objects with different surface properties and densities such as parasites, oocytes, micro structures or cells. To examine the differences in manipulating these kinds of objects, two simple experiments were performed to both pick and place an eyelash, then a single grain of pollen. In this case, manipulation was only carried out once to demonstrate the abilities of the micro gripper, rather than measuring pick and place time/attempts. Videos of both pollen and eyelash manipulation can be found in the Appendix A.

Pollen grains are sized within a range of approximately 3 μm–100 μm. Although this is slightly less than what has previously been manipulated with silicon tips, a 60-μm grain of pollen was still successfully picked and placed with a set of round tips (Figure 19). Pollen grains are also relatively spherical, which according to the previous experiment, makes round tips the most effective for manipulation.

It was also shown how a human eyelash can be manipulated with a set of square tips, where the average size for an eyelash is between 50 μm and 200 μm. In our case, the eyelash was 60 μm at the point of manipulation. Square tips were chosen in this case to provide a large gripping contact area for a firm grasp on the hair and to avoid damaging it (Figure 20).

### 3.4. Silicon Tips as Micro Tools

Lastly, it was demonstrated how different micro tips can also be used as tools at a micro scale; for example, a sharp tip can act as a cutting or puncturing device when we need to examine the interior of an object. This was demonstrated on a human eyelash, which was approximately 60 μm thick. A sharp tip was first aligned with the eyelash, gripped it and then cut through it (Figure 21).

### 3.5. Smaller Scale Manipulation

Thus far, all manipulation has been achieved using etched shaped silicon micro tips. These experiments have yielded important information on the functionality and efficiency of different tip shapes for a range of different micro objects. Up until now, successful manipulation has been demonstrated on both spherical and non-spherical micro objects, ranging from 60 μm—00 μm. Due to manufacturing limitations, manipulation with these tips has been limited to this size range. Many other micro objects such as micro machines or biological cells can extend well below this size range and often are found within 1 μm–20 μm; for example, the human red blood cell is approximately 8 μm in diameter. It is therefore important for the micro gripper to be also capable of manipulation within this smaller size range. Thus, for a long time in the literature, this size range has generally been avoided because of two key reasons. The first reason is manufacturing gripper tips at this scale is very difficult; however, the main reason is that adhesion forces begin to dominate at this scale, causing many things to stick together uncontrollably. This presents a challenge with manipulating at this scale, as micro grippers begin to require extremely small contact areas with micro objects. If the contact area is too large, such as the case with the silicon gripper tips, small objects become extremely difficult to release and stick to the gripper tips. Future work will include refining of the silicon gripper tips in order to reduce this contact area; however, for the time being, fine borosilicate glass end effectors were used. To demonstrate the capabilities and limitations of the micro gripper when it comes to this size range, a set of pick and place tasks was performed on silica micro beads sized within 1 μm–20 μm.

#### Manipulation Time

The purpose of using borosilicate glass end effectors as a micro gripper is to show that very small micro object manipulation between 1 μm and 20 μm is possible with the current compliant gripper system. At this scale, grasping each micro bead became very challenging as the size decreased. To show this, a sample of randomly-sized silica micro beads (within 1 μm–20 μm) was prepared, this time in a liquid environment. Experimenting in a liquid environment is preferred at this scale for contact manipulation, as it eliminates electrostatic and capillary forces [34,35]. A single bead for each size range (starting at 20 μm and decreasing evenly) was then selected for grasping. The time taken to achieve a stable grip was then recorded and repeated a total of 20 times for each bead size (Figure 22).

Across the range of 20 μm–12 μm, the average time was found to increase slightly in a relatively linear fashion from 9 s–32 s, respectively. This result was expected as a decrease in object size naturally requires a finer alignment and thus more user effort; however, this began to change as the bead size went below 12 μm. From this point, the average grasping time was shown to begin exponentially increasing. By the time the bead size had reduced to 6 μm, the average time had increased by more than nine times (303 s) since the 12 μm bead (32 s).

In order to pick and deposit these beads to a location in the sample, a simple method to prevent the bead from sticking to the gripper upon release was proposed. When the gripper tips were abruptly opened at high speed, the bead would often detach from the gripper tips and remain in position on the substrate floor. This was in part related to the end effectors being almost identical. When the tips opened at high speed, the adhesive force holding the bead to the end effector was less than the force required to accelerate the bead to the speed of release. Because the tips were reasonably similar, at the time of release, the accelerating forces from each end effector were equal and opposite. Because the adhesive force from either end effector was too weak to remain adhered to the bead while accelerating, the net force acting on the bead became zero, causing each end effector to break away at exactly the same time, subsequently causing the bead to remain stationary at the point it was once grasped. This can be seen happening in Figure 23, where the bead was seen to be released just a single camera frame after being grasped. The fast release of a 12-μm bead can be seen in the Appendix A.

Because the release speed is finite, there is a possibility that smaller beads may lack the necessary mass to break adhesion and therefore remain adhered to the tips, even under high velocity. To test this, a similar experiment was carried out; however, in this case; the success was recorded as either released or adhered to the gripper without any reference to the left or right end effector. Each bead size from 6 μm–20 μm was released at high speed over the course of 20 trials (Figure 24).

The results show that for bead sizes from 20 μm–12 μm, all were released successfully. Release success began to drop beyond this point with 10%, 20% and 60% of 10-μm, 8-μm and 6-μm beads adhering to the gripper, respectively.

## 4. Conclusions

To summarise the above results, it was found that certain gripper tip shapes can in fact increase the efficiency and effectiveness for manipulation of multi-shaped target objects. Firstly, when dealing with spherical objects, round tips showed the lowest pick and place time, as well as the lowest amount of average grasp attempts when compared to sharp and flat tips. This trend was also followed in the time taken data. Toothed tips were found as best for manipulating non-spherical objects for very similar reasons. The manipulation of a human eyelash and a pollen grain was demonstrated with silicon tips to show how micro beads are not the only option for the target object. This also showed the importance of tip selection and how it is possible to use certain tip shapes as tools for cutting/puncturing. As we have seen, biological objects can be easily damaged if gripped with a large force. In this study, selecting the correct gripper tip shape was used to avoid damaging the target object by increasing contact area. The minimum tip actuation of 0.1 μm and careful grasping process was also used to prevent accidental damage being caused to biological objects. This meant that objects were grasped in very small increments, avoiding over grasping and applying excessive force to the object. Furthermore, the biological objects manipulated in this study were relatively large; therefore, actual force sensing capabilities may be required in the future as biological object size decreases and they become more fragile.

One of the primary aims for this project was to develop a modular micro gripper with easily-replaceable gripper tips for different manipulation tasks. The initial idea was to achieve all manipulation of both large- and small-scale micro objects by using a vast variety of different etched silicon tips. Due to manufacturing limitations, it was impossible to create silicon tips with the required features small enough to manipulate objects below 60 μm. In order to manipulate cell-sized objects, glass end effectors were used. This allowed the gripper to maintain a modular design with both silicon-shaped tips and glass end effector capabilities.

For future work, improving the manufacturing of silicon tips and creating an easier method of manual alignment will be looked into. Currently, the gripper set up and usage relies heavily on the user. This can be time consuming and requires a certain level of operator skill. Because of this, we will investigate automating some of these processes in the future, such as tip alignment or pick and place operations.

## Figures and Tables

**Figure 1 micromachines-10-00313-f001:**
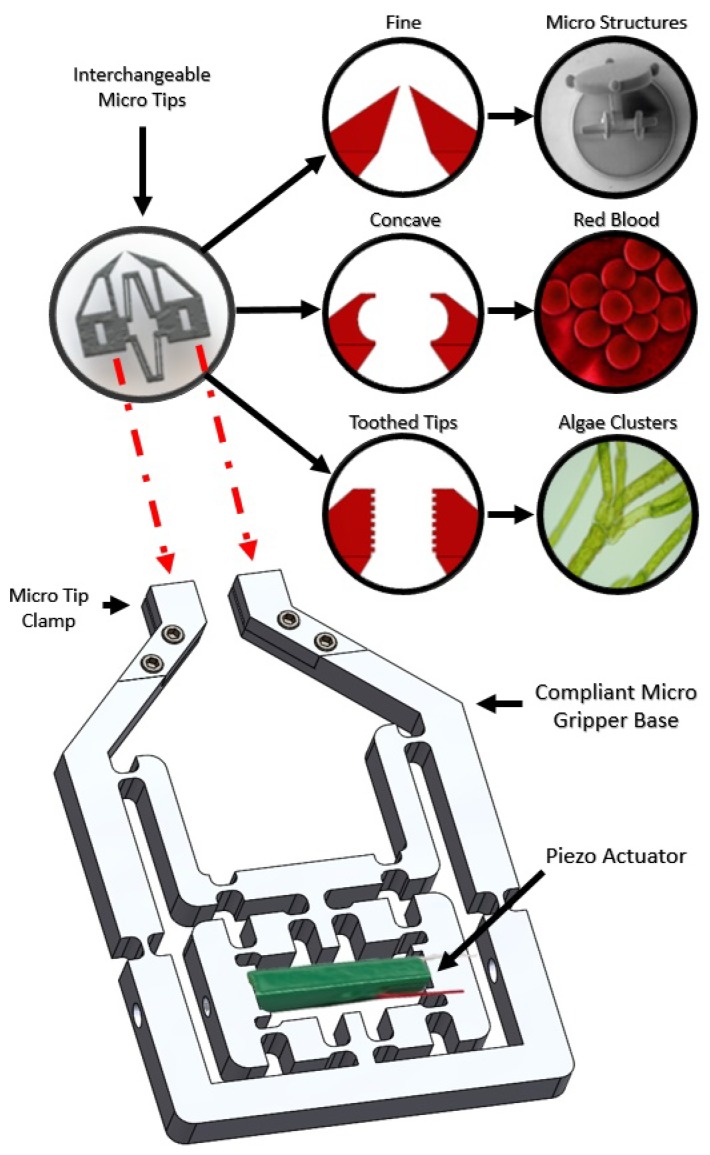
The modular micro gripper design showing the aluminium compliant base with piezo actuator. The silicon micro tip attachment can also be seen with three different tip geometries and example manipulation targets.

**Figure 2 micromachines-10-00313-f002:**
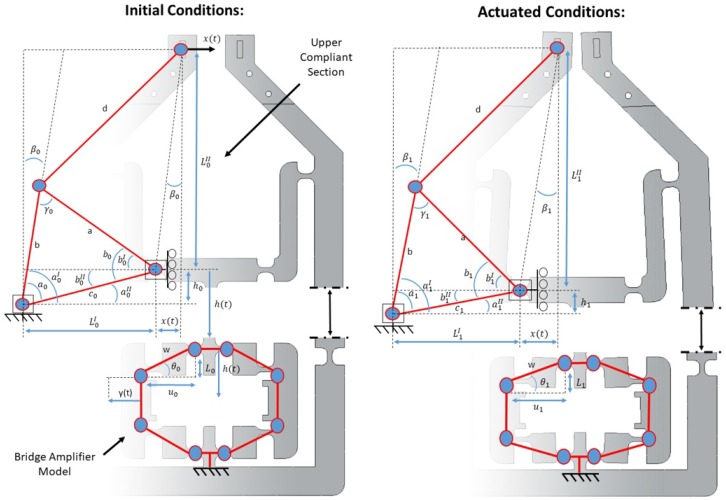
A detailed PRBM (psuedo rigid body model) representation of the compliant micro gripper base for kinematic modelling. Both before and after actuation conditions are shown. All design constants and variables relating to kinematic equations are labelled on the gripper.

**Figure 3 micromachines-10-00313-f003:**
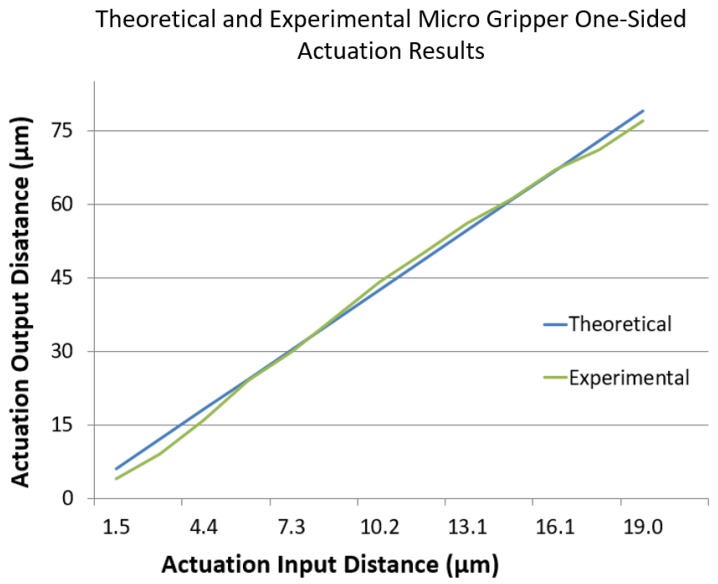
The measured/calculated theoretical and experimental output displacement for a single side of the micro gripper tips.

**Figure 4 micromachines-10-00313-f004:**
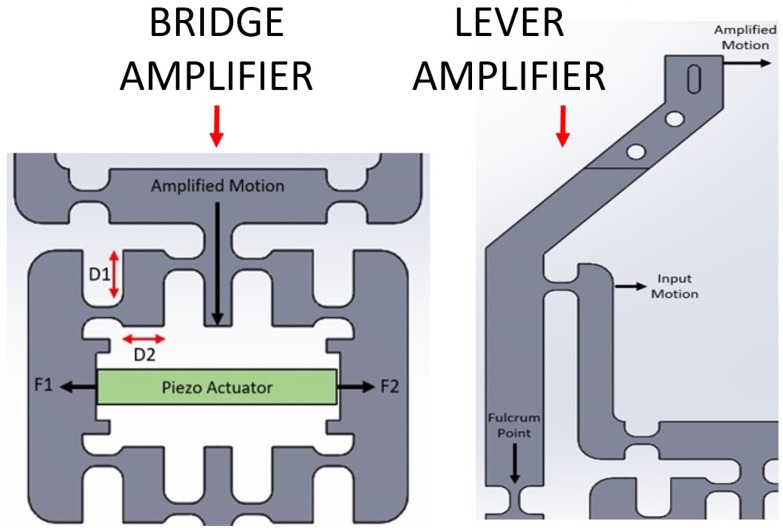
Computer-generated models showing the bridge amplifier with piezo actuator and the direction of input/output motion. The lever amplifier is also shown in a similar fashion.

**Figure 5 micromachines-10-00313-f005:**
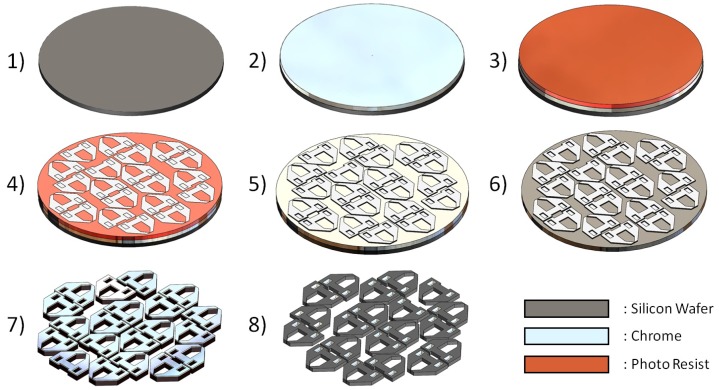
A sectional view showing the different layers involved in etching a silicon wafer (Stage 1: blank silicon wafer, Stage 2: silicon wafer with chrome layer, Stage 3: silicon wafer with chrome and photo resist layer, Stage 4: gripper pattern exposed into photo resist layer, Stage 5: gripper pattern in photo resist developed, Stage 6: chrome around developed gripper pattern removed, Stage 7: silicon etched away leaving only gripper tips with the chrome layer, Stage 8: chrome removed leaving only silicon gripper tips).

**Figure 6 micromachines-10-00313-f006:**
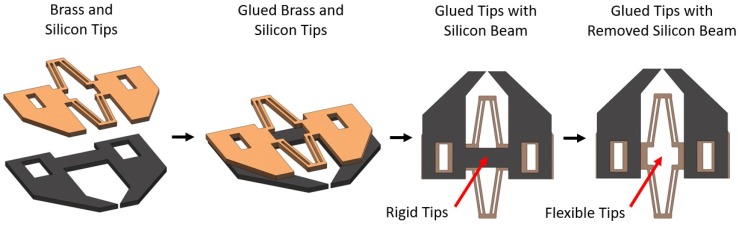
A computer-generated model showing the process of gluing the silicon micro gripper tips to the compliant brass tips.

**Figure 7 micromachines-10-00313-f007:**
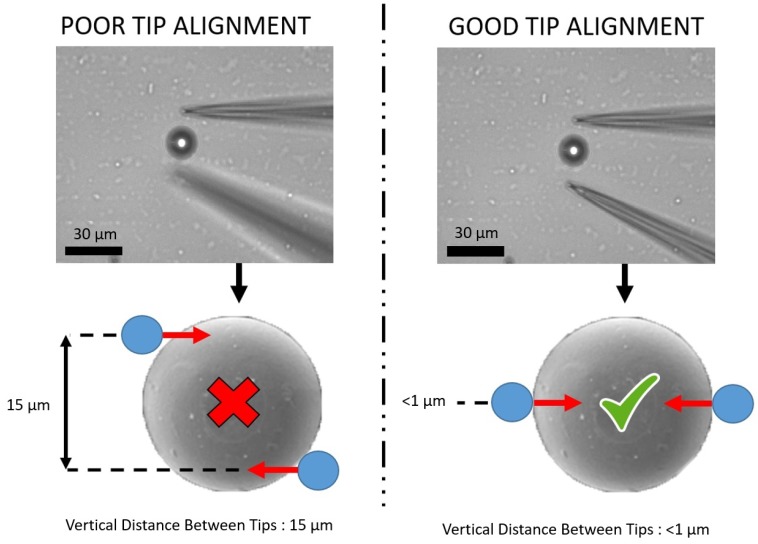
Diagram showing the importance of adequate tip alignment in grasping micro objects. Poor tip alignment with unsymmetrical micro tip object gripping is compared with good tip alignment, where gripper tips are perfectly aligned at the equator of the target object.

**Figure 8 micromachines-10-00313-f008:**
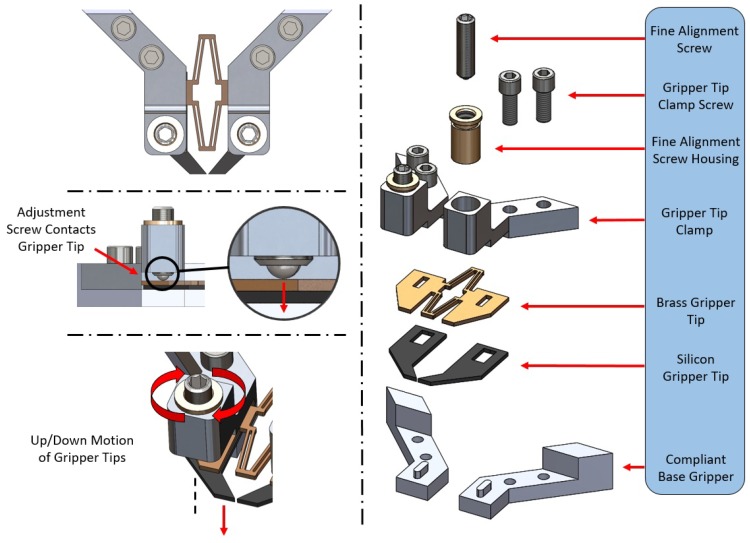
Computer-generated models showing a labelled exploded view of the fine adjustment screw alignment method for shaped silicon tips. The use and functionality are also shown.

**Figure 9 micromachines-10-00313-f009:**
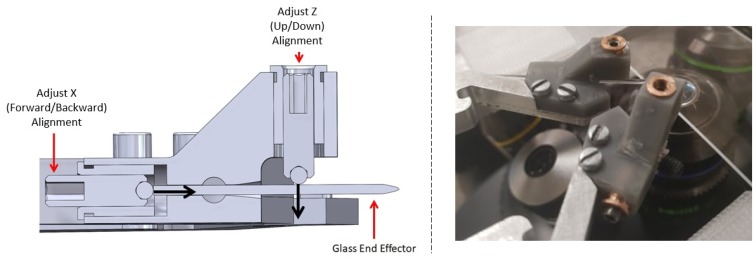
Computer-generated model showing a sectional view of the new glass end effector mounts used for alignment. Two fine adjustment screws are shown along with the direction of force used for alignment. An actual image of the setup in also included.

**Figure 10 micromachines-10-00313-f010:**
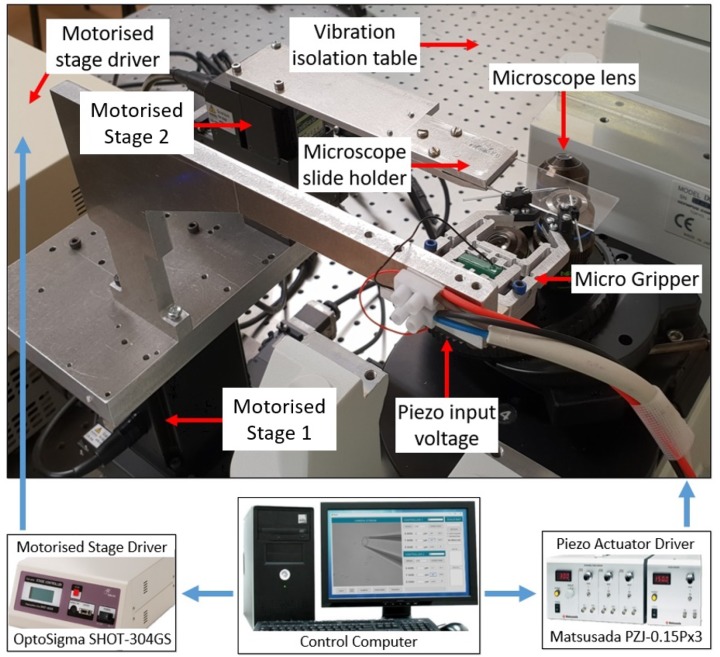
A diagram of the experimental setup for our entire micro gripper shown over a microscope. All key equipment related to the functionality, actuation and movement of the gripper are featured and labelled.

**Figure 11 micromachines-10-00313-f011:**
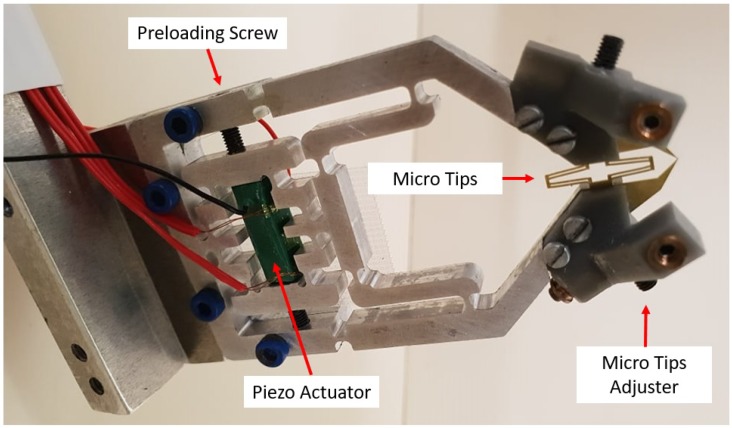
A close-up view of the compliant aluminium base gripper with a set of silicon/brass gripper tips attached. The piezo actuator, preloading method (screw) and also shaped silicon tip fine alignment mounts are shown.

**Figure 12 micromachines-10-00313-f012:**
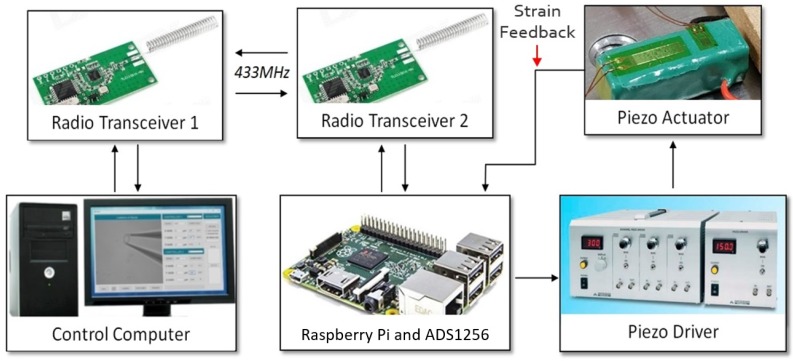
A diagram showing the methods and equipment used for transferring data around the micro gripper system, such as actuation commands or strain gauge feedback.

**Figure 13 micromachines-10-00313-f013:**
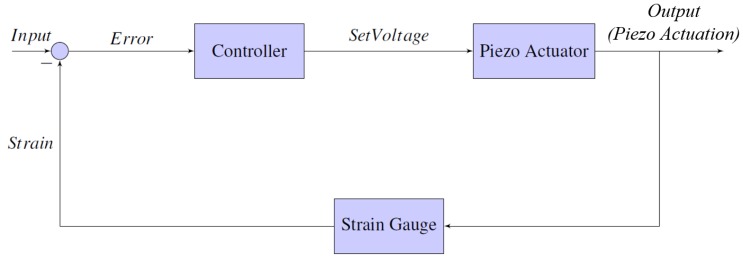
A simple closed-loop control diagram showing the actuation control system along with the strain gauge feedback direction.

**Figure 14 micromachines-10-00313-f014:**
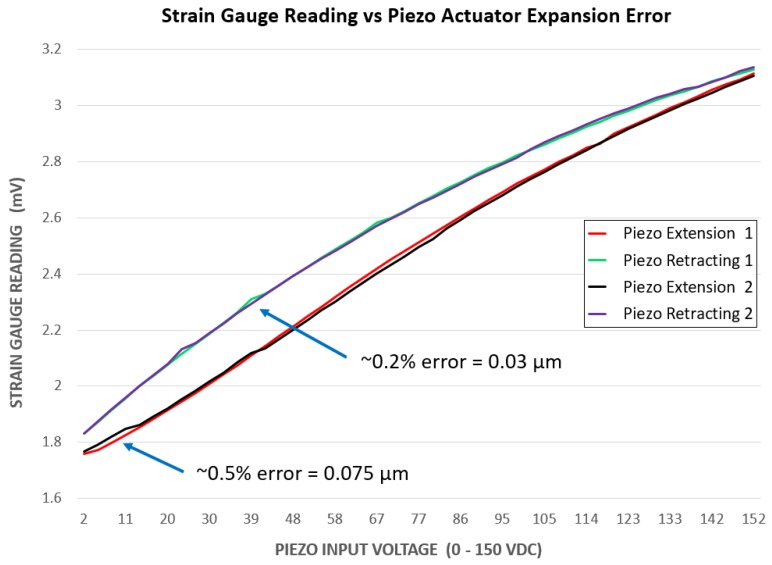
The maximum differences when comparing two expand/retract cycles shown side by side. Data shown represent recorded strain versus input piezo actuator voltage. The maximum error found across 10 expand/retract cycles is also shown and converted to piezo actuation.

**Figure 15 micromachines-10-00313-f015:**
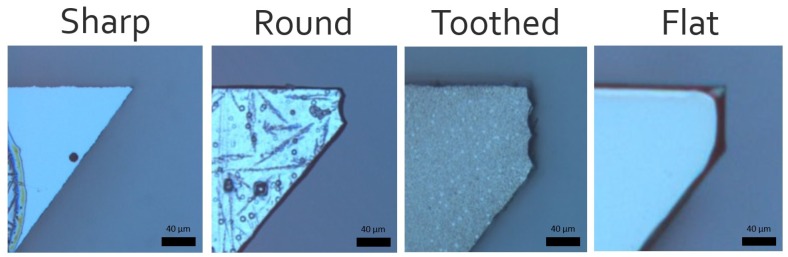
Four different silicon tip shapes (sharp, round, toothed and flat).

**Figure 16 micromachines-10-00313-f016:**
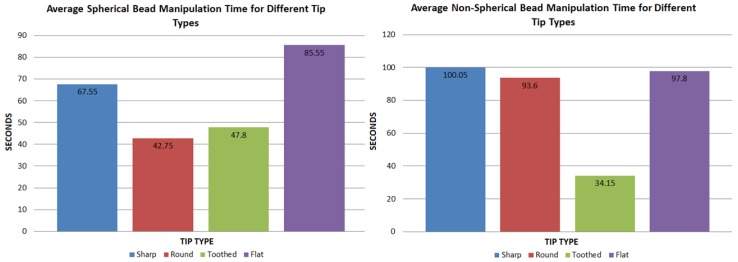
Average time taken to align, pick and move spherical and non-spherical micro beads correctly with silicon micro tips. The graph on the left shows results for spherical micro beads with the right side showing non-spherical results.

**Figure 17 micromachines-10-00313-f017:**
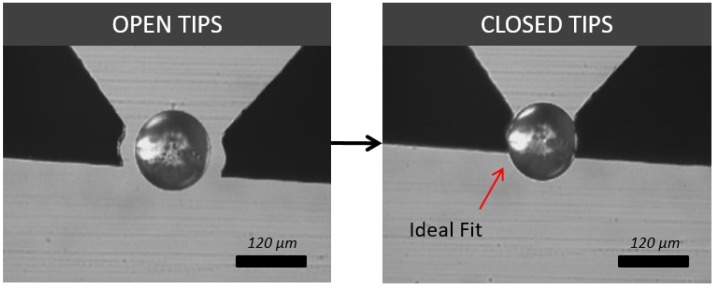
Round silicon tips showing image frames of before and after object grasping a 100-μm glass spherical micro bead.

**Figure 18 micromachines-10-00313-f018:**
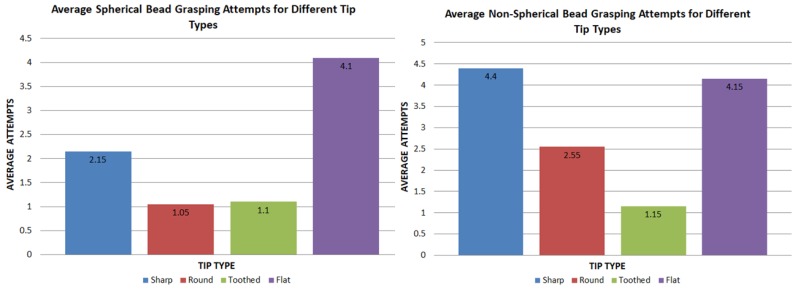
Average number of micro bead grasp attempts before achieving a firm grip. The graph on the left shows results for spherical micro beads with the right side showing non-spherical results.

**Figure 19 micromachines-10-00313-f019:**
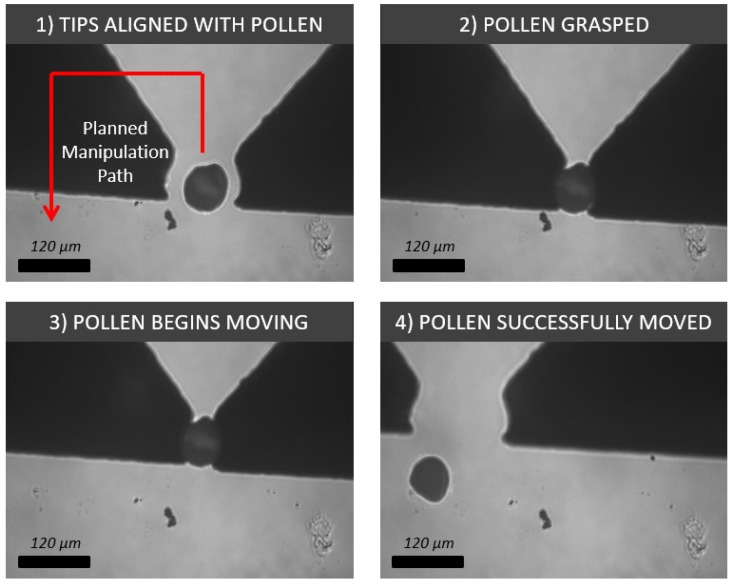
Process showing four image frames relating to the grasping, translating and placing of a 60-μm pollen grain with a pair of round silicon tips.

**Figure 20 micromachines-10-00313-f020:**
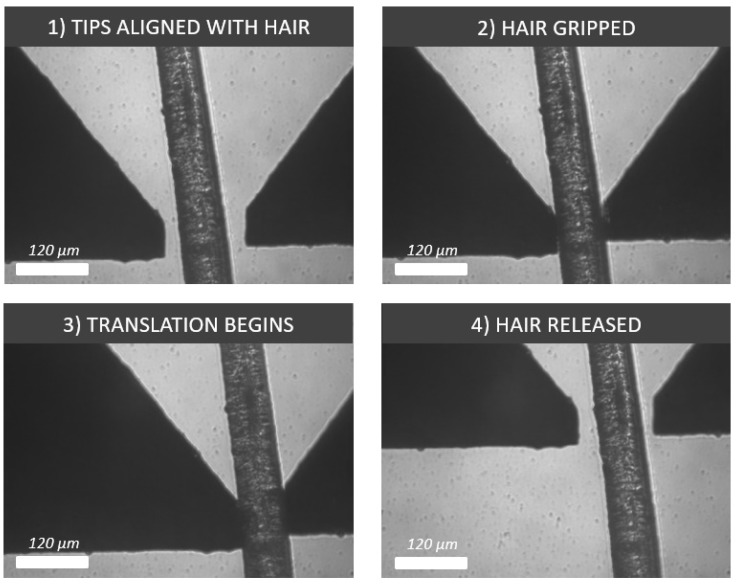
Process showing four image frames relating to the grasping, translating and placing of a 60 μm-thick human eyelash with flat silicon tips.

**Figure 21 micromachines-10-00313-f021:**
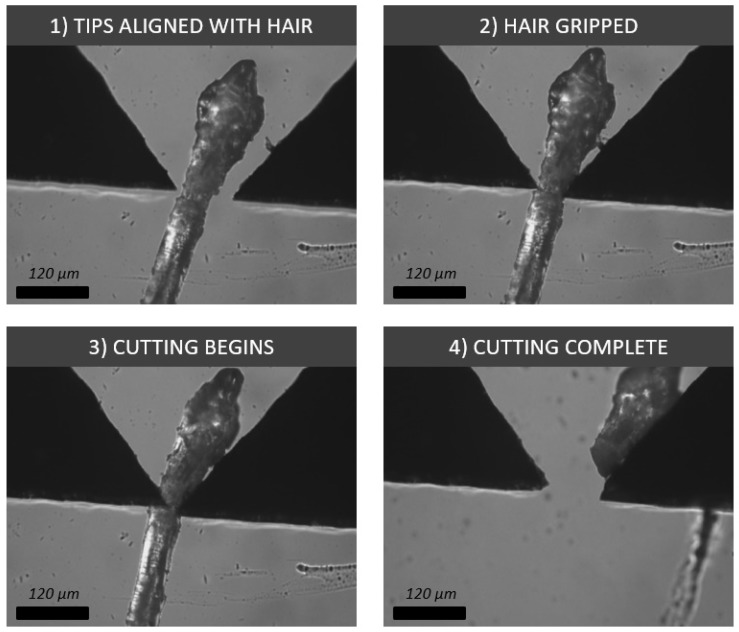
Process showing four image frames relating to the grasping and cutting of a 60 μm-thick human eyelash with sharp silicon tips.

**Figure 22 micromachines-10-00313-f022:**
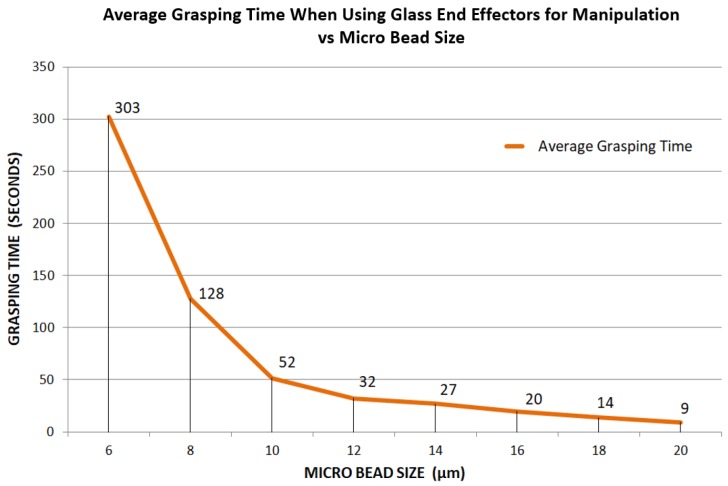
The average time taken to achieve a stable grip on different sizes of micro beads with glass end effectors.

**Figure 23 micromachines-10-00313-f023:**
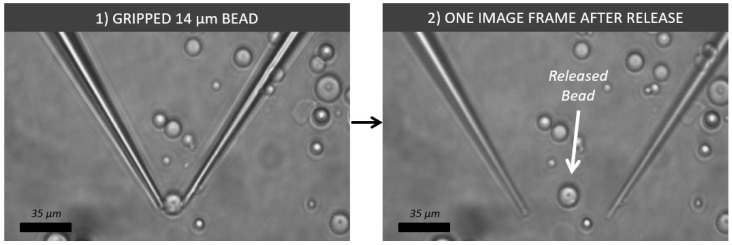
Image showing two image frames before and after high-speed release with a 14-μm micro bead. Grasping was performed by glass end effectors and was entirely released in a single camera frame later.

**Figure 24 micromachines-10-00313-f024:**
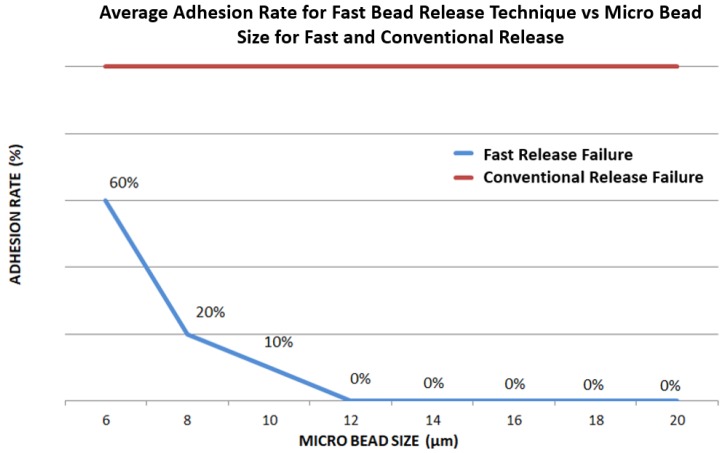
High-speed glass end effector micro bead release failure rate for different sizes of micro bead.

**Table 1 micromachines-10-00313-t001:** Compliant micro gripper design constants.

**Bridge Amplifier Constants**	
$u_{0}$	7 mm
$L_{0}$	3.5 mm
*w*	7.8 mm
**Upper Compliant Section Constants**	
*a*	17.8 mm
*b*	23.1 mm
$h_{0}$	6.3 mm
$L_{0}^{I}$	3.5 mm

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
