# Peer review of "Development of a Novel Modular Compliant Gripper for Manipulation of Micro Objects"

_micromachines, 2019, doi:10.3390/mi10050313_

Round 1

Reviewer 1 Report

The paper discusses the design, fabrication and experimental validation of a compliant gripper with interchangeable tips for micromanipulation.

The proposed concept is interesting addressing potentially relevant applications.

However, the paper requires major modifications prior to be published.

Specifically:

Section 1- Introduction

-         One of the advantages of non contact methods over contact manipulation is the fact that there is no force exchange between the tip and the object, and therefore it should be preferred when manipulating fragile or delicate objects, such as living cells. Moreover, living cells (such as red blood cells) usually require a wet environment, which amplifies the adhesion issues. Although the authors discuss some of the pros and cons of contact over non contact methods, more justification and supporting literature should be provided for this sentence: ‘Contact manipulation is generally preferred when working with living cells’ (lines 33-34).

-         The authors should also address the issue of sensing (and eventually controlling) forces between gripper and micro-parts.

Section 2 – Materials and methods

The authors should provide more information on the fabrication and assembly tolerances, specifically between the gripper body and the brass tips and between the brass tips and the silicon tips. It is not clear how the achievement of the correct alignment is verified.

Please check the title of Figure 3 (‘comparison of output displacement…’)

Section 3 – Experiments

Par. 3.1: The name of par. 3.1 – gripper control may be misleading. Please rename it (for example, gripper calibration). Moreover, please add some information about the calibration of the gripper tips. Ie: how the expansion/retraction errors of the actuator reflect on the tip errors? How often is the calibration procedure repeated?

Par.3.2 – method: please detail the type (shape, size, material) of the used ’20 randomly sized microbeads’ (line 256). Please also detail the grasping sequence. How were the poses of the gripper commanded? How repeatable was the experiment? How many trials have been performed?

Par. 3.3 Please detail the number of experiments also for pollen and eyelash. Line 302: please add dimensions of eyelash.

Author Response

Word file is uploaded.

Reviewer 2 Report

The manuscript titled "Development of a Novel Modular Compliant Gripper for Manipulation of Micro Objects," describes the development of a micro-gripper using piezoelectric actuation that is designed to grip small biological particles. The authors partially fabricate the device using microfabrication but they combine this with assembly of gripers and piezoelectric components. They validate their device using micro-beads and other materials focusing on the grippers shape. Overall the article is of interest to readers.

1. the paper would be better if the entire device was microfabricated, the authors mention gluing components using loctite. What is the alignment accuracy using this and how do you control that the loctitie doesn't flow until other surfaces. The resolution of the loctite cannot be very precise.

2. Micro-beads- what size micro beads did you use

3. What force does the actuators apply?

4. How fast did the grippers respond i would have thought the gripping time would be much lower than what it was.

5. how precise can they monitor displacement of the grippers to increase gripping force.

6. Are the grippers coated in any material or could you put a material on the grippers to improve adhesion.

Author Response

Word file is uploaded.

Reviewer 3 Report

The manuscript address the critical challenges in cell manipulation. Authors have rightly introduced the current challenges in this area. Adhesion has been a major shortcoming of the currently available devices.  Overall the manuscript is well written, and the design concept is well described through the geometrical representation. Following things, however, can be addressed to make the manuscript easily comprehendible.

Figures are not well explained. Please explain all figures in details in terms of the importance of the concept. 

Please comment on the robustness of the design in terms of usages. 

From Fig. 16 and 17, it is emphasized that manipulation efficiency is due to geometry which can be true if the applied force is the same in all condition. The toothed tip is efficient mainly due to geometry. Please explain the role of applied force through the actuator.

Author Response

Word file is uploaded.

Round 2

Reviewer 1 Report

The authors have addressed mostly of the previous comments.

Some additional comments are as follows:

Line 170: please rephrase the text: ‘because the brass and silicon tips are dimensionally identical’. This is ideally true but different manufacturing processes have different precision thus introducing manufacturing errors, as explained in lines 183-185.

Par. 2.5

Please specify the used vision system and its characteristics (resolution, magnification, …)..

Line 266: please add a semicolon between calibration and however.

Conclusions: The proposed gripper seems to mostly rely on the abilities of the human operator in aligning the tips, positioning, and grasping. Please discuss the extent to which the proposed microgripper could be used for automatic/semiautomatic operations.